# Predictors of Influenza Vaccination among Chinese Middle School Students Based on the Health Belief Model: A Mixed-Methods Study

**DOI:** 10.3390/vaccines10111802

**Published:** 2022-10-26

**Authors:** Yeerlin Asihaer, Mengyang Sun, Miao Li, Huidi Xiao, Nubiya Amaerjiang, Mengying Guan, Bipin Thapa, Yifei Hu

**Affiliations:** Department of Child, Adolescent Health and Maternal Care, School of Public Health, Capital Medical University, Beijing 100069, China

**Keywords:** influenza, vaccination, mixed-methods, health belief model, middle school students, hesitancy

## Abstract

Influenza vaccination rates among Chinese middle school students are low. This study aims to explore the influencing factors of vaccination among middle school students and promote vaccination. We conducted a mixed-methods study, integrating a questionnaire survey among 9145 middle school students in four cities in China and semi-structured interviews with 35 middle school students to understand their attitudes and perceptions toward vaccination based on the Health Belief Model. We found the overall vaccination rate was 38.2% (3493/9145), with students in Beijing, boarding at school, or senior high school showing higher values than their counterparts (*p* < 0.05). Multiple logistic regression results showed that non-boarding (OR = 0.46, 95%CI: 0.42–0.51) and perceived barriers (OR = 0.97, 95%CI: 0.96–0.98) were unfavorable factors for influenza vaccination, whereas perceived susceptibility (OR = 1.07, 95%CI: 1.05–1.08), perceived benefits (OR = 1.02, 95%CI: 1.01–1.04), cues to action (OR = 1.08, 95%CI: 1.05–1.11), and self-efficacy (OR = 1.04, 95%CI: 1.02–1.07) were facilitators. Qualitative results indicated that positive health beliefs, school, and the home environment contribute to vaccination. In conclusion, the influenza vaccination rate among middle school students remains low. The concerns about the safety and potential side effects of vaccines are the main barriers to vaccination, underscoring the need for strengthening communication, education, and information among students and their teachers/parents.

## 1. Introduction

Influenza (flu) is a global health threat [1], causing almost 4 million infections and 650 thousand deaths worldwide each year [2,3]. According to statistics on influenza prevalence among 13,329 unvaccinated persons in 32 randomized controlled trials of influenza vaccination worldwide, symptomatic influenza prevalence was 12.7% in children, 4.4% in adults, and 7.2% in elderly [4], indicating that children are at a higher risk of getting influenza. Poor air ventilation in classrooms and close contact between students make schools a high-risk site for influenza outbreaks [5,6,7]. More than 90% of the influenza outbreaks reported each year in China occur in schools and childcare institutions [8]. School-age children have the highest rate of influenza infection compared to other populations [9]. School-age children play an important role in bridging influenza spread in schools, families, and communities. Influenza epidemics can cause large numbers of school-age children to be absent from school, and parents to be absent from work [10].

Vaccination is the most effective strategy to prevent the spread of influenza [11]. During the 2017–2018 flu season, influenza vaccinations prevented 7.1 million illnesses and 41% of the expected hospitalizations of young children in the United States [12]. Large-scale centralized influenza vaccination can greatly reduce the risk of an influenza outbreak when the vaccine strain matches the circulating strain [13]. Additionally, even when the strain of influenza in the vaccine does not exactly match the one that is circulating, it can still effectively lower the risk [14]. In recent years, to increase the uptake of the influenza vaccine in adolescent populations, many programs have been launched. The school-located influenza vaccination (SLIV) program in the United States offers the best option for achieving high-immunization coverage in a short period of time [7]. The seasonal influenza program in England recommends children aged between 2 and 16 to be vaccinated with a live attenuated influenza vaccine (LAIV) [15]. However, many countries have reported low influenza vaccination coverage [16,17,18], which is considerably below the target of more than 75% suggested by the World Health Organization (WHO) [19].

The Health Belief Model (HBM), developed by social psychologists in the 1850s, is one of the most widely used models to study the relationship between health behaviors and health service utilization [20]. This model explains health behaviors in the following ways: perceived susceptibility to disease, perceived severity of disease, perceived benefits of an intervention, perceived barriers to an intervention, cues to action, and self-efficacy [21]. Research has shown that education and health beliefs influence adolescent vaccination decisions [22]. Moreover, previous studies have identified HBM’s components as important predictors of influenza vaccination [23,24,25], and its constructs provide a theoretical framework of the interview outline as well [26,27]. In order to inform health policy development to improve influenza vaccination rates and protect middle school students from severe outcomes of influenza infection, we explored the influencing factors of influenza vaccination among middle school students through a mixed-methods study based on HBM.

## 2. Materials and Methods

### 2.1. Study Design, Population, and Settings

We conducted a cross-sectional survey among middle school students in four Chinese cities (Beijing, Xi’an in Shaanxi province, Shenzhen in Guangdong province, and Anqing in Anhui province) in December 2020 by using convenient cluster sampling. We used the Wenjuanxing^®^ (Changsha Ranxing Information Technology Co., Ltd., Changsha, China) online survey tool (similar to SurveyMonkey^®^, San Mateo, CA, USA) to collect relevant information. We distributed 9463 questionnaires with informed consent. After excluding blank questionnaires (refusal to participate) and invalid questionnaires (incomplete responses), a total of 9145 questionnaires were analyzed (96.6%). Only Chinese schoolchildren were eligible, and only if they agreed to attend the survey, could read and comprehend the survey content, and were studying in ordinary middle schools rather than vocational schools.

Qualitative interviews were conducted from March to April 2021. A total of 35 middle school students from 4 schools in Dongcheng District, Beijing, were selected, using the purposive sampling method for semi-structured interviews, to investigate the influence of health belief factors on their vaccination decisions. Among them, 17 (48.6%) were boys, 18 (51.4%) were junior high school students, and 14 (40.0%) were boarding students. This study was approved by the Ethics Committee of Capital Medical University (Z2020SY116), and all participants signed the informed consent.

### 2.2. Questionnaire

The questionnaire consists of two parts. The first part contains demographic information, such as sex, grade, accommodation, region, and “whether to vaccinate against influenza this year (2020)”. The second part is a 20-item scale designed based on the HBM, using the Likert 5-level scoring method (Appendix A), including questions such as ”influenza vaccine will help me avoid missing class because of illness”, “I refused to get vaccinated because I was afraid of needles’ prick”, “TV news, WeChat subscriptions and other mass media say that the influenza vaccine is good for me and I should be vaccinated”, and “when I openly discuss vaccination with others, they may positively influence me”.

The Kaiser–Meyer–Olkin (KMO) value of the questionnaire was 0.853, and Bartlett’s Test of Sphericity’s *p*-value was <0.05, which is suitable for factor analysis. Exploratory factor analysis (EFA) with principal component analysis method using SAS 9.4 identified five dimensions: (1) perceived susceptibility, including three items, such as “I belong to a susceptible group” and “influenza vaccination is very important” (Cronbach’s alpha = 0.705); (2) perceived benefits, including four items, such as “influenza vaccine will help us avoid influenza” and “influenza vaccine will help save my parents’ time to care for me” (Cronbach’s alpha = 0.871); (3) perceived barriers, including seven items, such as “I am worried about adverse reactions after the influenza vaccination” and ” my classmates would observe me and I would feel embarrassed” (Cronbach’s alpha = 0.813); (4) cues to action, including three items, such as “according to the centers for disease control and prevention (CDC) public service announcement that getting the influenza vaccine is good for me and I should be vaccinated” and “The medical staff told me that the influenza vaccine is good for me and I should be vaccinated” (Cronbach’s alpha = 0.915); and (5) self-efficacy, including three items, such as “when I openly discuss vaccination with others, I will have a positive impact on the beliefs of others” and “I tend to listen to my parents about vaccinating or not” (Cronbach’s alpha = 0.713). The questionnaire has good reliability and the cumulative contribution rate of the five factors is 64%.

Confirmatory factor analysis (CFA) was performed using Amos 24. The results showed the goodness of fit index (GFI) = 0.955, adjusted goodness of fit index (AGFI) = 0.941, normed fit index (NFI) = 0.950, comparative fit index (CFI) = 0.952, incremental fit the index (IFI) = 0.952, and the root mean square error of approximation (RMSEA) = 0.051, indicating that the model fits well and has good construct validity (Figure 1).

### 2.3. Qualitative Interview

The qualitative interview adopted a self-designed interview outline based on HBM, including questions such as “Did you receive the influenza vaccine last year (2020)?”, “When the school organizes influenza vaccination, will you ask the peers around you whether they are going to be vaccinated?”, and “Do your parents support your influenza vaccination?”. Before we issued the formal survey, we solicitated opinions and comments from experts and middle school teachers on the rationality of the interview questions, the commonality of language descriptions, and the explanatory power. Further, we conducted pre-interviewing among two middle school students to form the final interview outline. The interviewers obtained information through face-to-face interviews, using group interviews, with 4 to 8 students in each group; a total of 7 groups were interviewed. Each interview lasted about 15–20 min.

### 2.4. Quality Control

The questionnaire was designed based on the literature on HBM and had been revised several times for good reliability and validity. The questionnaire was self-administrated by the middle school students themselves, who were able to fully understand the relevant questions and make choices based on their personal judgement. The online survey is conducted through Wenjuanxing^®^, a SurveyMonkey^®^-like online survey tool in China, avoiding the efforts such as training of field staff for information collection and therefore reducing measurement errors. The closed-ended online questionnaire allows for coding of answers in a patched process, avoiding errors in manual data entry and ensuring the accuracy and authenticity of the data during data merging and analysis.

The interview outline was optimized and revised several times based on HBM and the research objectives. With the consent of the interviewees, we made recordings and transcribed these verbatim, verified with the interview transcripts. The interview process was conducted in a quiet and undisturbed environment.

### 2.5. Statistical Analysis

Data were compiled in Microsoft Excel and exported to Statistical Analysis System software (V.9.4, SAS Institute Inc., Cary, NC, USA) for cleaning and analysis. Descriptive analysis was performed. Frequency tables with percentages were generated for demographic categorical variables, while the mean and standard deviation (SD) were calculated for scores of each dimension in the HBM. A Chi-square test was applied to determine the association between the different demographic variables and influenza vaccination, whereas an independent t-test was conducted to analyze whether the dimensions of the HBM had differences between the groups receiving an influenza vaccine or not. A multiple logistic regression was performed to identify the associated factors of influenza vaccination, where the scores of each dimension of HBM were considered as independent variables and whether or not to receive an influenza vaccine was the dependent variable, with sex and accommodation used as adjustment factors in the model. A two-tailed *p* < 0.05 was considered statistically significant.

The analysis of the interview data moved from initial coding to focused coding, through to theoretical coding, resulting in the production of core concepts and categories using NVivo 11 qualitative analysis software, which was completed in line with the process set out within constructivist grounded theory [28].

## 3. Results

### 3.1. Demographic Characteristics of the Participants

Among the 9145 participants, 3493 (38.2%) received the influenza vaccine. There was no difference between boys and girls in influenza vaccination rate (*p* > 0.05). There were differences in vaccination rates between stage of schooling, region, and accommodation, where vaccination rates among students in senior high school, living in Beijing, and boarding at school were higher than those of their counterparts (*p* < 0.05) (Table 1).

### 3.2. Univariate Analysis of Influenza Vaccination and Scores of Each Dimension of the HBM

The dimensions of the HBM presented differences between the groups receiving an influenza vaccine or not (*p* < 0.001). Students who received the vaccine were more likely to perceive influenza susceptibility, perceive the benefits of the vaccine, perceive the cues to action, and have higher self-efficacy. Students who did not receive the vaccine during the survey year (2020) were more likely to perceive the barriers of vaccination (Table 2).

### 3.3. Multiple Logistic Regression Analysis of Influencing Factors on Influenza Vaccination

The results showed that non-boarding (OR = 0.46, 95%CI: 0.42–0.51) and perceived barriers (OR = 0.97, 95%CI: 0.96–0.98) were unfavorable factors for influenza vaccination among middle school students, whereas perceived susceptibility (OR = 1.07, 95%CI: 1.05–1.08), perceived benefits (OR = 1.02, 95%CI: 1.01–1.04), cues to action (OR = 1.08, 95%CI: 1.05–1.11), and self-efficacy (OR = 1.04, 95%CI: 1.02–1.07) were facilitators of influenza vaccination (Table 3).

### 3.4. Qualitative Interview Analysis

This study analyzed the original interview data and extracted a total of 31 primary codes, such as “I don’t want to get sick”, “no one around has the flu”, “feeling troublesome”, and “afraid of prick or pain”. Based on those 31 primary codes, 10 categories were summarized, including “self-efficacy” and “perceived susceptibility”. Based on 10 categories, three main categories, namely, “health beliefs”, “school factors”, and “family factors”, were summarized.

#### 3.4.1. Health Beliefs

Perceived susceptibility: Results showed that many middle school students were aware of their susceptibility to influenza, and had the willingness to receive a vaccine: “I should get a flu vaccine this year because I took a two-week break due to a recent common cold”, “I am vulnerable to catch a cold, it’s very uncomfortable”, and “parents says if I get cold frequently, I will grow less than others”. However, some students did not realize their susceptibility to influenza, so they relaxed their vigilance: “I have never been sick”, and “No one at our school got the flu”. It can be seen that a lack of perceived susceptibility to influenza is a barrier for middle school students taking the influenza vaccination.

Perceived benefits: According to student reports, benefits included improved immunity, avoidance of absenteeism, and no need for physical education classes: “If I’ve had it before, I would have gained some immunity”, “Annual vaccination for so many years is very useful to avoid absentness in class”, “The last time you got flu shot, you said that you can skip gym class”, and “I don’t think it worth a risk for a long time of sore throat or even cough for a long time and I would rather take a shot”. It can be seen that the perceived benefits of vaccination can help drive more students to be vaccinated.

Perceived barriers: There were students who expressed their concerns about vaccine: “I was vaccinated when I was in elementary school, and I always had a fever, so I stopped getting vaccinated after then”, “I was vaccinated in my first year of junior high school, I was allergic, and the symptoms were obvious, so I won’t take the vaccine again”, “It’s useless”, “I heard people were pricked but they still caught a cold and had a fever”, and “I may be allergic over the special ingredients of the vaccine and parents told me”. Concerns about the side effects and efficacy of the influenza vaccine seem to be an important reason for preventing vaccination of middle school students.

Cues to action: In term of access to influenza vaccine information, most students reported “biology class”, and some others mentioned “hospital doctor”, “school doctor”, “parents”, “local public service announcements, anywhere in the traffic”. The more ways students know about access to the influenza vaccine, the more likely they are to get vaccinated.

Self-efficacy: When asked, “Do your parents listen to your opinion or do you abide with your parents’ opinion when it comes to influenza vaccination?”, some participants said that their parents would respect their own opinions: “Whether to get flu shot or not should depend on my current physical condition”, “Parents respect my opinion more”, and “It’s up to me to decide whether to get the flu vaccine or not”. Some students said that they listened to their parents’ decision: “Even if I want to listen to myself, I can’t get vaccinated unless they sign the informed consent”, and “I think parents should be informed about the benefits of this vaccine, because it is up to parents’ decision whether we will receive the vaccine or not.” It suggests that high self-efficacy drive influenza vaccination.

Attitudes towards vaccine: Students indicated mainly three different attitudes, namely, support, neutral, and resistance towards vaccines: “I have always got vaccinated every year”, “I don’t care”, and “I will not vaccinate”. A positive attitude can help with influenza vaccination.

#### 3.4.2. School Factors

During the interview, when asked, “When the school organizes influenza vaccination, will you ask the students around you if they are going to be vaccinated?”, most students answered “no” and “ask casually”. However, some students, especially boys, said that they “follow the crowd and do what others do”, “depend on the situation”, and “if more than half of the class were vaccinated, then I will vaccinate”. Furthermore, some students also used the WeChat group to discuss this tendency: “I can know who has been vaccinated and who did not via the parent WeChat group”, and “If more people get vaccinated, I will vaccinate”. In term of class teacher’s attitude towards the influenza vaccine, most students described it as “no interest” and “neutral”. Some students said that class teachers did not care about informed consent: “Just tell us not to forget to sign it”. However, due to the students’ admiration for their teacher, if the class teacher suggested everyone getting vaccinated, many students said, “as long as our class teacher recommends or supports, I will take a jab”.

It can be seen that the class/school atmosphere and class teacher’s cognition and attitude towards the influenza vaccine can directly or indirectly affect the students’ decision. A favorable atmosphere can drive more students to consider vaccination, thereby increasing the overall vaccination rate of the class.

#### 3.4.3. Family factors

Parents also showed three different attitudes, namely, support, neutral, and resistance towards vaccines: “definitely support, who would want their children to get sick”, “my parents said that I can get vaccinated or not”, “my mother didn’t let me vaccinate, and she never let me vaccinate since my childhood”, “mother would ask me how much proportion of students take shot and follow the majority and it is not a big deal”, and “mother would consult the my teachers and other parents but mostly she will support me vaccinating”. It can be seen that parents, as the guardians of the students, can directly influence and decide whether or not the students are vaccinated.

## 4. Discussion

Our study investigated the factors influencing influenza vaccination among middle school students in four cities through a mixed-methods study. The results showed that the self-reported influenza vaccination rate of middle school students in China was 38.2%, which is lower than the WHO’s proposed vaccination target of 75% [19] and that reported by another Chinese study (51%) [14]. Likewise, the vaccination rate is lower than the United States’ vaccination rate (58.6%) among children aged 6 months to 17 years in 2020–2021 [18] and vaccination rate (43.6%) among secondary school-aged children in the United Kingdom [29]. Nevertheless, it is encouraging that the vaccination rate in this study is higher than the vaccination rate of primary and middle school students (32.2%) in Beijing from 2016 to 2017 [30]. Studies have shown that concerns about vaccine side effects are an important reason for not receiving vaccine [31,32]. There are differences in susceptibility to adverse events, which varied by age; for instance, some adverse events, such as fever and the associated febrile seizures, are more common in children than adults [33]. According to the data of adverse events following immunization (AEFI) of seasonal influenza vaccines (InfV) administrated in China, during the 2015–2018 influenza season, the most frequently reported adverse reaction among all inactivated influenza vaccine (IIV) was fever, with a slightly higher incidence of hyperthermia (4.274/100,000 doses) in the children’s vaccine than the adults’ one (4.47 vs. 4.17/100,000 doses) [34]. We speculate that the low vaccination rate among middle school students is also due to concerns about adverse effects since there was a significant proportion of students in both the questionnaire survey and qualitative interviews who reported concerns about adverse reactions to influenza vaccination. It may be related to China’s scandals related to vaccine misconduct, leading to a loss of public confidence in vaccination [35].

The vaccination rate was significantly higher in Beijing than other three cities, which may be attributed to the provision of free vaccination services for primary and middle school students and the elderly since 2007 [36]. Furthermore, the influenza vaccination rate of boarding students was significantly higher than that of non-boarding students in this study. It may be related to the fact that boarding students spend more time living in schools and receive uniform education and life guidance, they may be less influenced by their parents, as parental influence is thought to be a factor causing hesitation about vaccines [37]. A study found that influenza vaccination rates increased with educational attainment in Japan, knowledge dissemination through education department have the potential to increase vaccine coverage [38], which could explain the higher vaccination rate among senior high school students in this study.

Our study found that students who received a vaccine this year showed greater levels of perceived susceptibility, benefits, cues to action, and self-efficacy in comparison to those who did not receive a vaccine this year. The findings of our study corroborate the results of a study on vaccination intentions in Hong Kong [39]. On the other hand, students who did not vaccinate this year reported a higher level of perceived barriers.

The results of this study are similar to findings on seasonal influenza vaccine uptake in China, both suggesting that high levels of perceived susceptibility and/or perceived vaccination benefits could drive influenza vaccination [40]. Summaries of the HBM-based theoretical framework found that the perceived barrier was significant in all studies in preventive health behavior studies [41]. Our study also showed that the perceived barrier was an unfavorable factor for influenza vaccination. Self-efficacy was a facilitator of healthy behaviors, consistent with many other studies [42,43]. Similar to the results of another HBM-based vaccine acceptance survey in Hong Kong [44], we found that cues to action was the strongest driver of vaccination promotion based on HBM.

Qualitative results suggested that HBM-based semi-structured interviews are effective in exploring the influencing factors of influenza vaccination. Results suggested that parents, as guardians of their children, often play a decisive role in getting their children vaccinated. This is consistent with other studies showing that parental influence was a significant hesitant factor contributing to low vaccination rates [45,46,47]. It suggests that, to overcome hesitancy, the safety of vaccines be enhanced, and effective health education be provided to both middle school students and their parents to address their concerns about the occurrence of adverse reactions. Another reason for low vaccination rates may be that the time of the interviews was in the middle or post-COVID-19 pandemic, where most people were vaccinated several times already; indeed, research suggested that influenza vaccination rates in the post-COVID-19 vaccine era be lower due to the perceived “vaccine saturation” phenomenon [48]. During the interviews, we found there were problems such as concerns about side effects of the vaccine, lack of perceived susceptibility and benefits, and low self-efficacy, which are consistent with the results of previous studies [49]. Hence, more research is needed to explore the factors that influence middle school students’ health beliefs about vaccination. The interview also revealed that class teachers, especially those who are affirmed and trusted by their students, play a favorable role in influencing their decision to vaccinate against influenza. We recommend that, in order to achieve a higher vaccination rate, schools should put more effort into mobilizing class teachers to advocate influenza vaccination.

The strengths of our study lie in the use of a combination of qualitative and quantitative research methods based on HBM. This mixed-methods study yielded more comprehensive and informative results through comprehensive data analysis. We also acknowledge several limitations of our study. First, this cross-sectional survey covered only four cities and may not represent the students of China entirely. Nevertheless, the four cities are located in eastern, southern, central, and northwestern China; hence, the characteristics of the students are well reflected. Second, the questionnaire did not cover the “perceived severity” of the dimension of HBM, since studies have shown it is not strongly associated with preventive health behaviors [41]. Third, whether the middle school students received the influenza vaccination was self-reported and recall bias is unavoidable; however, it is rare for middle school students to forget an event of receiving a jab within one year.

## 5. Conclusions

This study investigated the factors influencing influenza vaccination among middle school students through a mixed-methods approach based on the theory of HBM. We found that the influenza vaccination rate of middle school students was mainly affected by school stage, region, accommodation, perceived susceptibility, perceived benefits, perceived barriers, self-efficacy, and cues to action. Our findings could inform policy makers and public service announcements to address concerns about vaccine safety and build a favorable school atmosphere, so as to increase the vaccination rate of middle school students.

## Figures and Tables

**Figure 1 vaccines-10-01802-f001:**
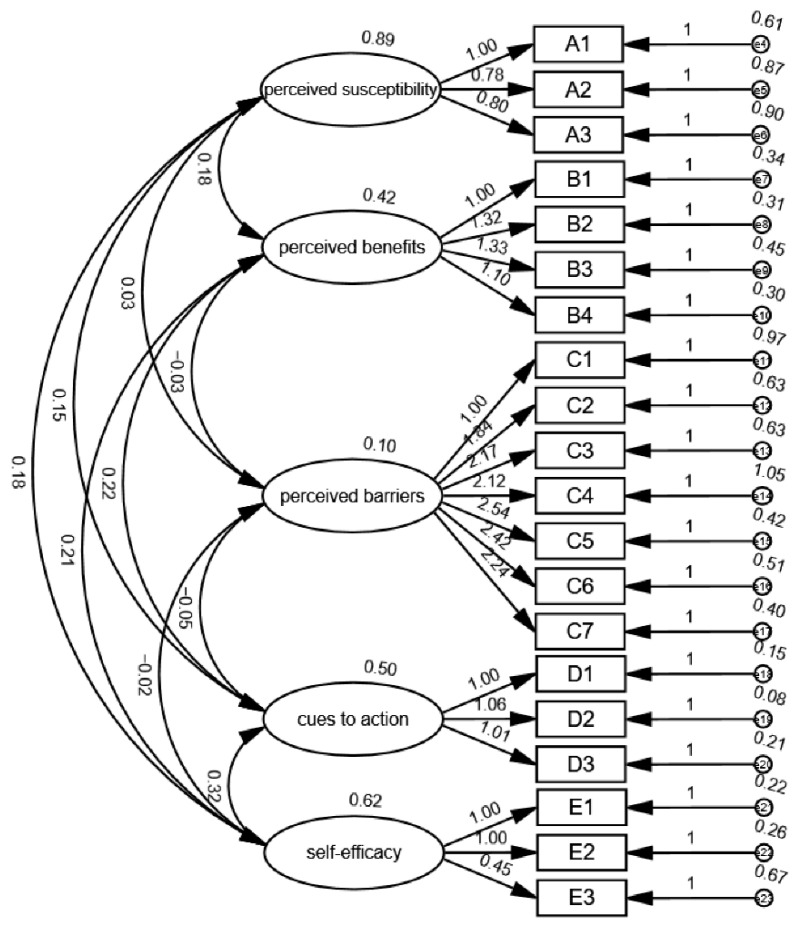
Confirmatory factor analysis structural equation plot.

**Table 1 vaccines-10-01802-t001:** Comparison of the influenza vaccination rates among middle school students with different demographic characteristics.

Characteristics	Number of Students (%) (*n* = 9145)	Vaccinated Students (%) (*n* = 3493)	Not Vaccinated Students (%) (*n* = 5652)	*p*
Sex				0.31
Boy	4570(50.0)	1769(38.7)	2801(61.3)	
Girl	4575(50.0)	1724(37.7)	2851(62.3)	
Stage of schooling				0.041
Junior high	5702(62.4)	2132(37.4)	3570(62.6)	
Senior high	3443(37.6)	1361(39.5)	2082(60.5)	
Region				<0.001
Beijing	4999(54.7)	2598(52.0)	2401(48.0)	
Other three cities	4146(45.3)	895(21.6)	3251(78.4)	
Accommodation				<0.001
Boarding	1822(19.9)	969(53.2)	853(46.8)	
Non-boarding	7323(80.1)	2524(34.5)	4799(65.5)	

**Table 2 vaccines-10-01802-t002:** Comparison of influenza vaccination among middle school students with different HBM dimensions scores (*n* = 9145).

Dimensions	Influenza Vaccination		*p*
	Yes (Mean ± SD *)	No (Mean ± SD *)	
Perceived susceptibility	9.45 ± 2.95	8.75 ± 2.81	<0.001
Perceived benefits	16.19 ± 3.18	15.47 ± 3.36	<0.001
Perceived barriers	16.21 ± 5.12	16.94 ± 4.94	<0.001
Cues to action	12.09 ± 2.15	11.43 ± 2.29	<0.001
Self-efficacy	11.26 ± 2.19	10.73 ± 2.17	<0.001

* SD: standard deviation.

**Table 3 vaccines-10-01802-t003:** Multiple logistic regression analysis of the influencing factors on influenza vaccination among middle school students (*n* = 9145).

Factors	OR (95%CI) ^a^	*p*	OR (95%CI) ^b^	*p*
Sex				
Boy	ref.	-	ref.	-
Girl	0.96(0.88,1.04)	0.31	0.95(0.87,1.04)	0.26
Accommodation				
Boarding	ref.	-	ref.	-
Non-boarding	0.46(0.42,0.51)	<0.001	0.46(0.42,0.51)	<0.001
Perceived susceptibility	1.07(1.06,1.09)	<0.001	1.07(1.05,1.08)	<0.001
Perceived benefits	1.02(1.00,1.03)	0.017	1.02(1.01,1.04)	0.008
Perceived barriers	0.97(0.97,0.98)	<0.001	0.97(0.96,0.98)	<0.001
Cues to action	1.08(1.05,1.10)	<0.001	1.08(1.05,1.11)	<0.001
Self-efficacy	1.04(1.02,1.07)	<0.001	1.04(1.02,1.07)	0.001

^a^ Univariate logistic model; ^b^ Multiple logistic model adjusted for sex and accommodation of students; OR: odds ratio, CI: confidence interval.

## Data Availability

The data that support the findings of this study are not publicly available but are available from the corresponding author on reasonable request.

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
