# Peer review of "Predictors of Influenza Vaccination among Chinese Middle School Students Based on the Health Belief Model: A Mixed-Methods Study"

_vaccines, 2022, doi:10.3390/vaccines10111802_

Round 1

Reviewer 1 Report

Comments to the authors:

Summary: The paper evaluates the factors influencing influenza vaccination among middle school students through a mixed-methods approach based on the HBM. The authors conducted a mixed-methods study integrating a questionnaire survey among 9145 middle school students in four cities in China. They showed that the influenza vaccination rate of middle school students was affected by school stage, region, accommodation, perceived susceptibility, perceived benefits, perceived barriers, self-efficacy and cues to action. I recommend the publication of this article after consideration of minor comments below:

1.     Introduction: Similar study has been published. Authors should cite the following paper. Low coverage of Influenza vaccination among Chinese children aged 12-23 months: Prevalence and associated factors. PLoS One. 2018; 13(10)

2.     Authors should check for punctuation errors.

3.     Can the authors clarify if the survey included only Chinese students? Can they comment on middle school students from other ethnic groups?

4.     4. Can the authors elaborate on why they have chosen only four cities.

Reviewer 2 Report

This study aims to explore the influencing factors of influenza vaccination among middle school students in order to promote vaccination.

The introduction section should be improved. Particularly, I suggest improving the aims of the study. Particularly, in an experimental study, the aims should be clearly written: indeed, the goals should be SMART (Specific, Measurable, Attainable, Relevant, and Timely). Moreover, considering that the target referred to middle school students, data about the statistics on influenza prevalence among unvaccinated should be referred to this age. The introduction section should focus on this particular population.

The material and methods should be improved. Several important data should be inserted. The average age (considering that the first part of the questionnaire is self-administered), the income of families, etc.. 

The results section summarized the main data.

The discussion should be improved. In particular, the authors should discuss several important points. First of all, this study is conducted in a particular age group. Usually, the risks of adverse effects related to the vaccination could be relevant compared to the beneficial effect of flu vaccination in this age group. This aspect should be discussed. Moreover, another important consideration concerns the time of the interview: it is located in the first post-COVID era; the data could be influenced by this aspect. Please, discuss it. Furthermore, another important limitation of this study is related to the self-administration of the questionnaire. How did the authors ascertain the reliability of the answers? Please, discuss this important aspect.

Minor points:

-Following the authors' instructions (https://www.mdpi.com/journal/vaccines/instructions), "The abstract should be a total of about 200 words maximum". Please, check it.

Round 2

Reviewer 2 Report

Following the authors' suggestions, the authors have improved their manuscript.